# Mitigation of *Salmonella* in Ground Pork Products through Gland Removal in Pork Trimmings

**DOI:** 10.3390/foods12203802

**Published:** 2023-10-17

**Authors:** Reagan L. Jiménez, Mindy M. Brashears, Rossy Bueno López, David A. Vargas, Marcos X. Sanchez-Plata

**Affiliations:** International Center for Food Industry Excellence, Department of Animal and Food Sciences, Texas Tech University, Lubbock, TX 79409, USAmindy.brashears@ttu.edu (M.M.B.);

**Keywords:** lymph node, risk assessment, quantification, *Salmonella*

## Abstract

Bio-mapping studies conducted in pork harvest and fabrication facilities have indicated that *Salmonella* is prevalent and mitigations are needed to reduce the pathogen in trim and ground products. *Salmonella* can be isolated from the lymph nodes and can cause contamination in comminuted pork products. The objective of this study was to determine if physically removing topical and internal lymph nodes in pork products prior to grinding would result in the mitigation of *Salmonella* and a reduction in indicators in the final ground/comminuted products. In total, three treatment groups were assigned in a commercial pork processing facility as follows: (1) untreated control, (2) topical (surface) glands removed before grinding, and (3) topical, jowl, and internal lymph nodes and glands removed before grinding. Indicator microorganisms were determined using the BioMérieux TEMPO^®^ system and the quantification of *Salmonella* was performed using the BAX^®^ System Real-Time *Salmonella* SalQuant^®^ methodology. The removal of lymph nodes located on the topical and internal surfaces and in the jowl significantly (*p* < 0.05) reduced the presence of *Salmonella* and also reduced the presence of indicator organisms according to this study. Briefly, 2.5-Log CFU/sample of *Salmonella* was initially observed in the trim samples, and the ground samples contained 3.8-Log CFU/sample of *Salmonella*. The total numbers were reduced to less than 1-Log CFU/sample in both trim and ground products. This study indicates a need for lymph node mitigation strategies beginning prior to harvest, in order to prevent contamination in further-processed pork products.

## 1. Introduction

Pork is now number two in meat consumption globally. As of 2020, 106.3 million tons of pork are consumed annually around the world [1]. The National Pork Producers Council has reported that in the United States (U.S.), more than 2.2 million metric tons of pork and pork-related products are exported annually [2]. The value of these exports is approximately USD 7.7 billion [3]. In the United States, there were 129.9 million pigs slaughtered in the U.S. in 2019, all of which entered the food supply chain. The United States Department of Agriculture (USDA) reports that pork is consumed as fresh cuts of meat such as chops, ribs, roasts, or hams and the remaining is consumed in the form of processed pork such as sausages, hot dogs, and bacon [4].

*Salmonella* is often reported as the leading cause of foodborne illness in U.S. populations [5]. Every year in the U.S., non-typhoidal *Salmonella* is responsible for approximately 1,027,561 cases, 19,336 hospitalizations, and 378 deaths [6], resulting in USD 3.7 billion worth of costs to the United States economy [7]. The Interagency Food Safety Analytics Collaboration (IFSAC) reported these attributions in October 2021 for 2019 in collaboration with data from the Centers for Disease Control and Prevention (CDC), the U.S. Food and Drug Administration (FDA), and the United States Department of Agriculture’s Food Safety and Inspection Service (USDA-FSIS). They reported that 75.9% of *Salmonella* cases were attributed to the following seven major food categories: chicken, fruits, pork, seeded vegetables, other produce, turkey, and eggs. In total, 12.8% of these cases were attributed to pork [8]. These data indicate that pork is the second highest contributor to salmonellosis cases from FSIS-regulated products, and the third highest contributor from all food products. 

Previous studies have biomapped microorganisms in pork processing facilities to determine the prevalence of *Salmonella*, generic *E. coli*, and other indicator organisms at various stages of the pork processing chain. These studies reported that pathogen and indicator organism prevalence were reduced throughout the processing line, but increased again in trim and further processing stages, creating a U-shaped curve of the biomapped organism [4]. The facility in this study was operating as a HACCP-Based Inspection Models Project (HIMP) facility prior to the New Swine Inspection System (NSIS) for over 20 years. *Salmonella* prevalence, as determined via the previously developed baseline biomapping study, is 19% within the entire facility. Out of the 650 samples collected during the baseline biomapping study, 125 samples were positive at 24 h under prevalence testing. However, only 47 out of the 125 were positive at the 6 h SalQuant^®^ time point, meaning that only these select samples were quantifiable [4]. Two samples were selected in this facility for sample collection: boneless picnic trim and final product brick sausage. The previous study concluded that the selected testing locations for further processed products had a 24% prevalence of *Salmonella* positives. This increase in bacterial prevalence in further processed products when compared to the facility average indicates a need for intervention strategies at the later pork processing stages. To mitigate *Salmonella* in trimmings and ground pork products, novel intervention strategies must be studied for efficacy and cost effectivity for industry applications because to date, there are few interventions available.

Lymph nodes are embedded within various muscle tissue groups, and these glands are used to filter lymphatic fluid during the lifespan of the animal. According to recent research, lymph nodes in both pork and beef have been shown to carry a high load of *Salmonella* after harvest [9]. Currently, pork trim and ground products are processed with the following lymph nodes and glands included in the muscle tissue. 

Topical
○This group includes superficial popliteal lymph nodes located in the back legs, as well as superficial inguinal lymph nodes located in the fat on the medio-ventral surface of the hind leg. This group may also include various accessory glands and notably includes subiliac nodes.
Jowl○The pork jowl contains 2–3 salivary glands, such as the paratoid and submaxillary glands, and several lymph nodes down the jowl, cheeks, and neck.Internal
○The se deep tissue lymph nodes are located in the fat in the crease between the semitendinosus (eye of round) and gastrocnemius (knuckle). This group also includes the gluteal and ischiadic lymph nodes located on the sarcotuberal ligament. This group also contains various glands from the loin region such as the vesicular and bulbourethral glands.

The removal of glands and lymph nodes from boneless picnic hams prior to grinding for sausage production is a possible means to reduce *Salmonella* prevalence in further processed pork products, but little data exist on node/gland removal as a mitigation strategy. The objective of this study was to determine if physically removing topical and internal lymph nodes in pork products prior to grinding would result in the mitigation of *Salmonella* and a reduction in indicators in the final ground/comminuted products, thus identifying the targeted glands and lymph nodes as sources of *Salmonella* contamination in further processed pork products. In the future, the authors of this study will aim to further investigate mitigation strategies to reduce *Salmonella* prevalence in these structures.

## 2. Materials and Methods

### 2.1. Sample Collection and Treatment

Pork trim and ground sausage samples were collected online from a large-scale USDA-inspected hog processing facility. The facility where samples were collected is a large hog processing facility that is currently processing approximately 10,400 per head per day and 1250 per head per hour on average, located in the United States. This facility is USDA-inspected and is currently operating under the New Swine Inspection System (NSIS), as proposed in 2019. To operate under the NSIS, the processing plant was required to implement specific worker safety measures, including an agreement with a workers’ union to represent their employees [10]. These safety measures allow the processing plant to work at higher line speeds under supervision and regular testing requirements to ensure safety and quality standards while increasing productivity. According to a constituent update published by the USDA FSIS in 2021, facilities that are not operating under the time-limited trial of the NSIS can only process up to 1106 per head per hour. Picnic trim comprises lean muscle and fat trimmings that come from the picnic shoulder of the carcass. The shoulder is a muscle of the hog that typically is tougher due to the amount of work put on the muscle. This trim, along with formulated fat and spices, is most often used as the meat component for ground sausage products within this facility. Ground sausage samples are taken after final packing in vacuum brick packages with easy-open seals. This sausage is formulated in accordance with the company’s recipe independent of the treatment group. In total, 15 samples were taken from each sampling location for each treatment group (n = 90) per repetition. The entire study was replicated 5 times over a period of 4 months to account for natural variation and seasonality (N = 450). The individual sampling dates (23 March 2022, 29 March 2022, 4 April 2022, 20 May 2022, 1 June 2022, and 16 June 2022) were used to account for seasonality and natural variation within the product. FSIS directive number 65-20 was utilized to collect trim and ground samples. Protocols for whole pork cuts (intact and non-intact) and comminuted pork aseptic grab sample not in final packaging were followed as they were for the nationwide sampling program. As written in the protocol, a 375 g sample of fresh, not frozen, raw pork was collected and placed into a single sterile Whirl-Pak bag (Millipore Sigma, Burlington, MA, USA). A 1 lb portion of ground pork sausage was collected in its final packaging from the facility. To conduct raw material sample collection, 15 samples of each of the three treatment groups were taken after gland removal on each day of sampling with five replications.

Samples were shipped overnight to the food microbiology laboratory at Texas Tech located in the International Center for Food Industry Excellence (ICFIE) after immediate chilling. Samples were processed and evaluated for Enterobacteriaceae (EB), Aerobic Count Bacteria (AC), and *Salmonella* concentrations. Additionally, retainer samples were kept at the plant for further evaluation. Chilled pork carcasses were fabricated through a standard process. As the shoulder passed through the production line, the foot, jowl, and neck bone were removed. The butt was then separated from the picnic to generate a bone in, skin on picnic. The picnic was then transferred to the boning department through conveyors and equipment. The treatment of sample groups 2 and 3 was performed on belts, conveyors, and other equipment that only received gland-free products during processing to avoid cross-contamination. Between treatments, the belts and equipment were thoroughly sanitized by trained personnel in order to reduce any potential cross-contamination. All picnic trim samples were collected from a singular vat per treatment group. The treatment groups are defined as follows:Treatment 1—standard trim on boneless picnic hams—control
○This treatment included standard trim without removing additional glands/defects. Skin, bone, meat, trim and inedible tissue were removed. The skin, meat, bone, trim, and inedible tissue were collected and weighed for the vat to obtain yield information. The vat of picnics was identified and they were sent for sausage production.
Treatment 2—retail trim on boneless picnic hams
○Here, additional trimming was required; blood clots, and all surface/exposed glands were removed regardless of color. Skin, bone, meat, trim, and inedible tissue were removed. The skin, meat, bone, trim, and inedible tissue were collected and weighed for the vat to obtain yield information. The vat of picnics was identified and they were sent for sausage production.Treatment 3—export trim on boneless picnic hams
○In this treatment, the bones in picnics were removed for standard trimming with the addition of removing exposed glands and surface blood clots regardless of size and color. Glands associated with the jowl and glands inside the boneless picnic were removed, as were skin, bone, meat, trim and inedible tissues. The skin, meat, bone, trim, and inedible tissues were collected and weighed to obtain the vat for yield information. The vat of picnics was identified and they were sent for sausage production.

These treatment groups were consistent for both the trim and group samples.

### 2.2. Processing Methodology

Upon arrival at Texas Tech University, the samples were evaluated for any leaking, damage, or potential temperature abuse. A 50 g aliquot of the sampled pork cut was weighed into a filtered Whirl-Pak bag (55 oz). A 200 mL portion of 45 °C (pre-warmed) of BAX^®^ MP media (Hygiena™, Camarillo, CA, USA) was added to the sample bag. A stomacher (Model 400 Circulator, Seward, West Sussex, UK) was used to homogenize the trim samples at 230 rpm for 30 s. The processing of the ground pork sausage samples followed a similar protocol. First, 50 g of the product was weighed into a Whirl-Pak bag (55 oz, filtered) and a pre-warmed 200 mL aliquot (45 °C) of BAX^®^ MP media was added. Ground pork samples were then homogenized in a stomacher for 1 min at 230 rpm. A 30 mL aliquot of the homogenate from the primary Whirl-Pak bag was aseptically transferred into another filtered Whirl-Pak bag (24 oz) using a disposable serological pipette (Fisher Scientific, Foods 2022, 11, 2580 5 of 20 Waltham, MA, USA). To the aliquot in this bag, a 30 mL portion of BAX^®^ MP Media containing 1 mL of Quant solution (Hygiena™, Camarillo, CA, USA) was added to the 30 mL pure homogenized sample. An additional 10 mL aliquot of each sample type (both ground and trim) was transferred using a serological pipette into sterile tubes to enumerate indicator microorganisms, which was conducted before the samples were incubated for *Salmonella* enumeration and prevalence. The utilized processing methodology was adapted from a previous study conducted to biomap *Salmonella* and indicator organisms at each step of the pork processing line [4]. This study was replicated 5 times over a period of four months to account for the natural seasonality and variability of the pathogens (N = 450).

### 2.3. Microbial Analyses

Indicator bacteria were enumerated using the TEMPO^®^ system (BioMérieux, Paris, France). The method of the Association of Official Agricultural Chemists (AOAC) 121204 was used to enumerate AC. Briefly, the method calls for the incubation of TEMPO cards for 22–28 h at 35 ± 1 °C. For EB, enumeration cards were incubated for 22 h at 35 °C. For the TEMPO enumeration of indicator organisms, the original sample was diluted to a 1/20 dilution in all sample and indicator types. To prepare this dilution, 3 mL of water and 1 mL of the sample rinsate was added to a dehydrated media vial. This dilution was then filled into the correlating TEMPO card for each indicator, EB or AC, and incubated according to the directions for each organism. Once incubated appropriately, the cards were read using TEMPO Reader. Results were converted into Log10 values for interpretation and evaluation. 

In order to quantify Salmonella levels in the collected pork samples, trim samples were placed into a 42 °C incubator for 6 h, and ground pork sausage samples were incubated for a 7 h period for quantification. Following the AOAC 081201 protocol, after incubation for 7 h, *Salmonella* was enumerated using the BAX^®^ System SalQuant^®^ (Hygiena, Camarillo, CA, USA).The AOAC Level 2 validation of BAX^®^ System Real-Time Polymerase Chain Reaction (RT-PCR) Assay for *Salmonella* and BAX^®^ system SalQuant^®^ (Certification No. 081201) followed. An aliquot of each sample was taken for the enumeration protocol, and then the original sample bags (containing the homogenate) were put back into the incubator at 42 °C for 18–24 h to detect any *Salmonella* that might have been present but below the limits of quantification.. 

The sample preparation protocol for the BAX^®^ System Real-Time PCR Assay for *Salmonella* has 3 stages for the workflow: preparation, lysis, and PCR. The first stage, sample preparation, consisted of preparing the lysis reagent in accordance with the provided protocol and thermal blocks that were pre-heated to 37 °C and 95 °C. The lysis step was completed by transferring 5 µL of the sample to cluster tubes, and then a heating step at 37 °C for 20 min was carried out. Additionally, a subsequent heating step was conducted at 95 °C for 10 min. Upon the completion of the steps, samples were cooled for 5 min. The PCR stage of this protocol involved hydrating PCR tables with 30 µL of the lysate and running the BAX^®^ Q7 thermocycler with the appropriate assay parameters. 

### 2.4. Data Analysis 

To evaluate the microbiological results, all data were analyzed using R (Version 4.1.2) statistical software. Each treatment was compared to the control. Counts of indicator organisms were converted into LogCFU/g and *Salmonella* counts were reported as LogCFU/sample. A one-way ANOVA (analysis of variance) was performed on the data, which compared the pathogen counts from each of the treatment groups, followed by pairwise multiple comparison *t*-tests, and adjusted via the Bonferroni method. *p*-values of 0.05 or less were used to determine significant differences. 

Data were arranged into boxplots, with a horizontal line within the box to represent the median of the data. The lower (0.25) and upper (0.75) quartiles are represented by the top and bottom lines of the box. The upper and lower lines represent 1.5 times the interquartile range. The dots present on the plots represent the actual collected data points. For each matrix, boxes indicated with different letters are reported as significantly different between treatments according to *t*-test analysis at *p*-value < 0.05.

## 3. Results

The LogCFU/g counts of AC indicate significant differences between the control samples, the retail trim, and the export trim. Enterobacteriaceae results show a statistically significant difference among the control trim and both treatment groups, but there was not a significant difference between EB counts obtained from retail trim and export trim. *Salmonella* counts were recorded and presented in Log_10_CFU/sample using a 50 g sample basis. Trim samples had overall higher counts of both indicator organisms and *Salmonella* for all treatment groups when compared to ground samples from the same treatment groups. Indicator organisms, especially EB, show a wide range of variance for each set of samples, which indicates an overall need for better process control methods within this facility in order to reduce the variation. 

### 3.1. Detection and Quantification of Salmonella 

In total, 72/450 samples tested positive for Salmonella (16%). Table 1 shows *Salmonella* prevalence from each treatment group. In total, 38 samples (52.7%) were suitable for enumeration with the majority being detected from treatment group 1, the control group. The breakdown of collected positives from each treatment group and matrix can be found in Table 1. 

Overall, 72 of the 450 samples tested positive for *Salmonella* in the prevalence assay after 24 h of enrichment (n = 72). Of these, 43 samples were part of the control group, 22 of the positive samples were from the retail trim (topical gland removal only) group, and 7 positives were a part of the export trim category, which had the topical, jowl, and internal glands removed. Overall, more *Salmonella* positives were detected in ground samples as opposed to trim samples, and this was displayed across each treatment group.

*Salmonella* counts were very low in the majority of the samples as analyzed on a per gram basis. Therefore, for visualization purposes, all data were transformed into LogCFU/sample, which is equivalent to LogCFU/50 g, to facilitate data interpretation. The limit of quantification (LOQ) for SalQuant^®^ on pork trim and ground pork was 0.1 CFU/mL and 0.1 CFU/g or 0.70 LogCFU/sample. When samples were negative for quantification, they were reported as 50% of the LOQ (0.35 LogCFU/sample). Samples that were not quantifiable or detectable were reported as 0 LogCFU/sample but these are not reflected on Figure 1 as including these values would have altered the mean values of the positive samples. 

Of the 450 collected samples across five replications, 38 samples were positive at the SalQuant^®^ time point for quantification (n = 38). The recorded quantitative values are displayed in Figure 1. The control group averaged at 2.5 Log CFU/sample and 3.8 Log CFU/sample of *Salmonella* in ground and trim samples, respectively. The export trim group held the lowest average of *Salmonella* counts for both matrices at less than 1 Log CFU/sample. There were statistical differences among each of the three treatment groups for both detection and quantification methodologies. Of the 31 quantifiable samples, 3 were from the export trim, 12 were from retail trim, and 16 were from the control trim. The mean of each sample point was used to determine significant differences between the sample groups. There was a significant difference (*p* < 0.05) between the control samples and each of the treatment groups. However, there was not a significant difference between the retail trim and export trim treatment groups. 

### 3.2. Enumeration of Enterobacteriaceae and Aerobic Count Bacteria

AC counts are described in Figure 2. Total ACs were statistically compared across mean values for each treatment group and matrix.

As shown in Figure 1, there was a statistical difference (*p* < 0.05) among the treatment groups for both matrices with the control having the highest, retail having a smaller difference than that of the control but a higher difference than that of the export, and export having the lowest. Export trim, which was composed of boneless picnic trim with the topical, jowl, and internal glands removed, had the lowest average AC counts for aerobic plate counts for both the ground and trim matrices. The lowering of AC counts indicates a reduction in overall microbial activity within the samples collected from each treatment group as the glands were removed. 

EB counts, as detailed in Figure 3, were compared in terms of mean value for each treatment group and matrix.

Enterobacteriaceae counts were determined using the TEMPO system and converted into Log_10_CFU/g values. While the range for each treatment group remains wide across treatments, the median value for each group decreases as the lymph nodes and glands are removed. The means of each sample point were used to determine significant differences among treatments. There was a statistical difference (*p* < 0.05) among EB counts collected from the control group, and counts collected from each of the treatment groups. Unlike the case for AC counts, there was not a statistical difference between treatment retail trim and export trim. a reduction in EB microorganisms is beneficial to the product as it indicates a lower amount of potential pathogenic presence. 

The pattern of the detected and quantified *Salmonella* correlates closely with the pattern of the EB and AC indicator organisms measured within this study. While these values correlate in pattern, there is not an exact ratio between the relationships. Since the organisms follow similar patterns, indicator organisms can be observed to suggest the presence of *Salmonella* in pork products; however, the observation of indicator organisms cannot be utilized in place of *Salmonella* testing in this particular operation. The similar pattern followed by *Salmonella* and the indicator organisms additionally suggests that the removal of lymph nodes affected the organisms directionally.

## 4. Discussion

The quantification of *Salmonella* in pork samples from commercial industry establishments may be limited because of pathogen recovery, due to pathogen stress caused by the processing environment, and the application of antimicrobial interventions. The quantification techniques utilized within this project have been validated to recover pathogens from positive samples as a result of a recovery stage. The use of short enrichment steps strengthens the quantification data via the recovery of injured cells [11,12]. Additionally, *Salmonella* quantification may offer an opportunity to make risk-based and data-driven decisions based on the prevalence and overall concentration at specific processing seps in the process, rather than the presence or absence of the pathogen [13]. The quantification of pathogens can benefit the pork processing industry as indicated by the results of this study, which provides evidence for novel uses of emerging pathogen detection technologies. The utilization of a rapid PCR-based enumeration methods for *Salmonella*, in conjunction with the enumeration of indicators, provides the pork industry with a tool to make data-driven decisions to reduce pathogenic prevalence in trim and further processed pork products and to mitigate the risk of public health of foodborne illness.

Furthermore, the results of this study indicate that the removal of topical lymph nodes and glands from boneless picnic trim was an effective method for reducing *Salmonella* in boneless picnic pork trim and ground sausage products in this operation. Furthermore, the results of this study indicate that the removal of topical, jowl, and internal glands and lymph nodes further reduced the prevalence of *Salmonella* and other indicator organisms in boneless picnic trim and ground sausage when compared to that with the removal of topical glands and lymph nodes alone. It is important to note that in this study, strict sanitary measures were used in node and gland removal and the amount of time taken to remove the nodes and glands was significant; thus, the same procedure may be difficult to implement in commercial operations. There could be a risk of cross contamination if the nodes are not carefully removed and proper sanitation protocols are not implemented. If this method is chosen to mitigate *Salmonella*, effective and efficient methods for gland and lymph node removal should be determined and developed before the implementation of these strategies within the industry. It is also critical to understand which nodes contributed to the most reduction. Nodes were not isolated and it could be a single node or combination that resulted in the reductions. Finally, the serotype and the pathogenicity of the *Salmonella* was not determined. In order to make an impact on public heath, serotypes of the highest concern that are related to human illnesses should be considered. Additional mitigation strategies should also be observed, such as pre-harvest strategies to prevent node contamination, or chemical/physical applications of interventions for reducing pathogenic prevalence within further processed pork products. This study clearly establishes that lymph nodes contribute to *Salmonella* presence in ground products. Additionally, the results of this study indicate that the lymph nodes and glands identified within this study and removed from the treatment groups were probable sources of *Salmonella* contamination in further processed pork products. The authors of this study further recommend the identification of mitigation strategies to reduce *Salmonella* prevalence within the lymph nodes as the removal of these structures in a large-scale pork processing facility is not currently feasible due to the high labor demand of the removal process. 

Pork facilities also face proposed performance standards from the FSIS that must be met. While not yet implemented, the current proposed performance standards determine that the “pass or fail” status of a processing plant are based on the total number of *Salmonella* positives taken from samples over a 52-week rolling window. However, the positives are based on the detection methodology that determines if any amount of *Salmonella* is present within the tested sample. Pork processing plants are currently being tested for the presence or absence of *Salmonella* alone and not for quantification, determination, serotype, or pathogenicity. It is currently estimated that the infectious dose of *Salmonella* is relatively high when compared to that of other pathogens, being estimated to be between 10^5^ and 10^6^ cells [14]. The quantifiable *Salmonella* positives detected within this study showed levels far below 10^6^ CFU. This suggests that pathogenic loads of the product might be below or far below the number of cells that would cause human infection in a healthy adult upon the consumption of a fully cooked product. Therefore, this data could be used to inform future decisions regarding performance standards and their dependency on quantification-based methodologies for *Salmonella* testing rather than presence–absence alone in order to make more informed decisions about the safety of a product. Implementing quantification methodologies within the industry for pathogenic testing may provide further insights and data to make risk-based public health decisions.

In conclusion, the results of this study indicate that the lymph nodes and glands targeted and removed in the treatment groups of this study could be identified as probable sources of *Salmonella* in further processed pork products. This information can be used as foundational support for further research to be conducted on the implementation of mitigation strategies to reduce pathogenic prevalence within these structures.

## Figures and Tables

**Figure 1 foods-12-03802-f001:**
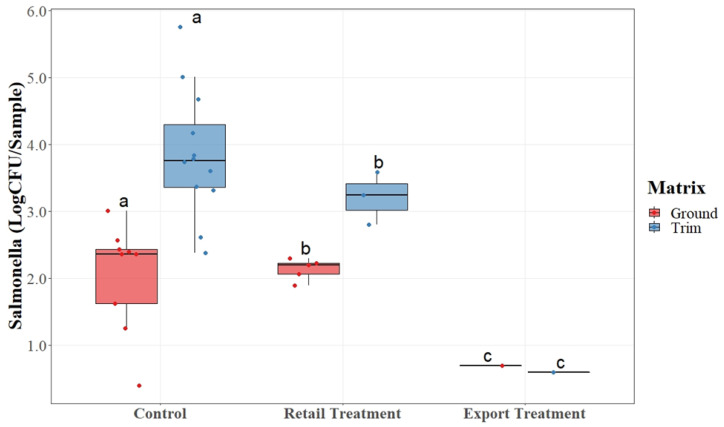
*Salmonella* quantification as determined via SalQuant^®^ on pork trimmings and ground product subjected to gland and node removal. Horizontal lines are the median; upper and lower quartiles are represented by the top and bottom lines of the box. Each dot represents a data point. a, b, c: for each matrix, boxes with different letters are significantly different among treatments according to a *t*-test analysis at *p*-value < 0.05.

**Figure 2 foods-12-03802-f002:**
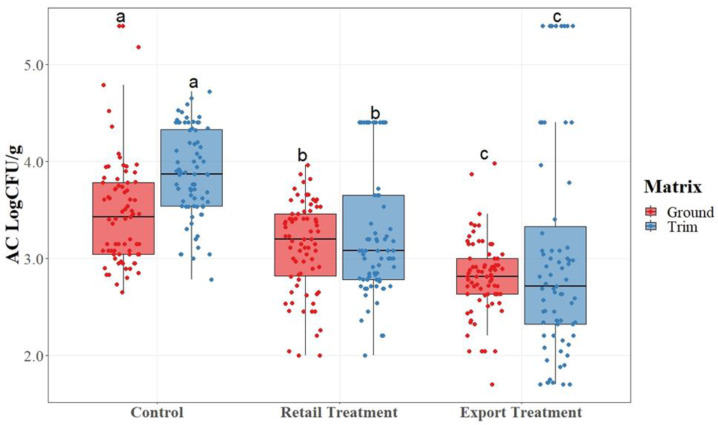
Aerobic plate count (AC) of ground pork and pork trimmings collected over a 4-month period (n = 450) with and without gland removal. The mean of each sample point was utilized to determine significant differences among treatment groups. a, b, c: for each matrix, boxes with different letters are significantly different among treatments ac-cording to a *t*-test analysis at *p*-value < 0.05.

**Figure 3 foods-12-03802-f003:**
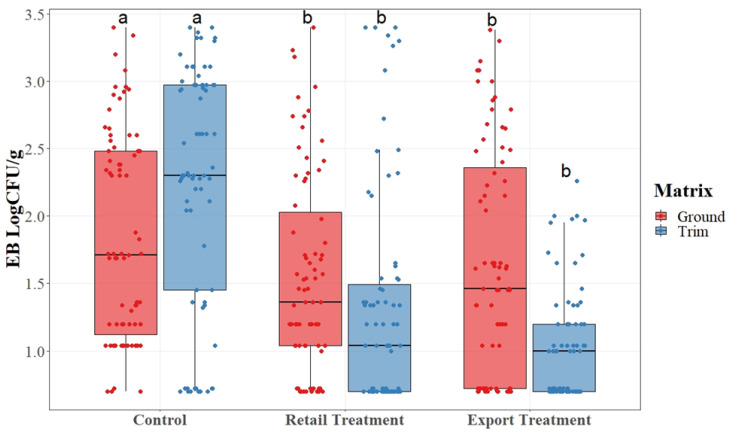
Enterobacteriaceae (EB) results for pork trim and ground pork collected from a commercial pork facility with and without lymph node removal. a, b, c: for each matrix, boxes with different letters are significantly different among treatments ac-cording to a *t*-test analysis at *p*-value < 0.05.

**Table 1 foods-12-03802-t001:** Percentage of samples testing positive for *Salmonella* upon SalQuant^®^ analysis using the BAX^®^ system using commercially obtained ground pork and pork trimmings with and without node removal.

Product	Control (%)	Retail Trim (%)	Export Trim(%)
Ground Pork with Seasonings	30.5 n = 22	19.4 n = 14	5.6 n = 4
Boneless Picnic Meat with Different Trim Levels	29.2 n = 21	11.1 n = 8	4.2 n = 3
Total	59.7% n = 43	30.5% n = 22	9.8% n = 7

## Data Availability

Data are available upon request from the corresponding author. The data are not publicly available due to privacy from the pork processing partner which allowed the project to be conducted within their facility.

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
