# Peer review of "Mitigation of Salmonella in Ground Pork Products through Gland Removal in Pork Trimmings"

_foods, 2023, doi:10.3390/foods12203802_

Round 1

Reviewer 1 Report

This paper addresses an important issue in pork processing: the presence of Salmonella and indicator organisms, particularly in trim and ground products. The study focuses on the impact of removing glands and lymph nodes from pork to mitigate this issue.

The methodology employed in this study is well-structured and employs established techniques for assessing microbial presence. The use of the BioMérieux TEMPO® system and BAX® System Real-Time Salmonella SalQuant™ methodology adds credibility to the research.

The division of samples into three treatment groups, including an untreated control, allows for a robust comparison of outcomes. This controlled experimental design strengthens the validity of the findings.

The results are presented clearly and include specific quantitative data, such as Log CFU/sample counts. This numerical representation offers a transparent view of the reduction in Salmonella prevalence as a result of gland and lymph node removal.

The paper's aim is clearly stated: it aims to ascertain whether removing glands and lymph nodes from pork leads to a reduction in Salmonella and indicator organisms. This precise research question forms a robust basis for the study.

The methodology relies on FSIS Directive Number 65–20 for sample collection procedures. Offering a concise overview or referencing the critical elements of this directive would aid readers in comprehending the thoroughness and standardization of the sampling process.

Nevertheless, a pertinent question arises: is this practically feasible? If so, what steps would need to be taken to implement it?

In my view, the introduction falls short in providing comprehensive support for the study.

Reviewer 2 Report

The work by Jimnez et al comprises the reduction in Salmonella prevalence by removal of glands and lymph nodes. By considering the volume of pork produced and traded annually, the control of salmonella in pork products is very crucial for food safety. The manuscript has some important findings regarding salmonella control in the pork processing. The authors further validated the concept of removal of lymph nodes and glands in the pork to significantly control the salmonella count.

The language is easy to understand and clear. The hypothesis is clear.

General comments:

Lot of acronyms are used without their full names. Please use full name before acronyms at the when the acronyms are used first time.

Other observations are as follows-

       i.          L18-19: please check the p levels. For significant p<0.05

     ii.          Keywords:  numbering in keywords? Plz check and remove

   iii.          L32: plz correct  106.3 million tons consumed annually per capita? Please rewrite.

   iv.          Rather I would suggest to delete first para. Authors may emphasize this at Line 48 by stating that giving the data of the volume of pork produced in the USA etc.

     v.          In introduction, Authors have detailed the lymph nodes removed, I would appreciate if author provide information of the glands removed in this study also.

   vi.          Methodology: Appropriate

  vii.          L223: a,b,c?

viii.          Results: Good presentation of results

   ix.          Discussion: well supported by references

     x.          Conclusion and the final remark are lacking.
